# A Meta-Analysis of Wearable Contact Lenses for Medical Applications: Role of Electrospun Fiber for Drug Delivery

**DOI:** 10.3390/polym14010185

**Published:** 2022-01-03

**Authors:** Hamed Hosseinian, Samira Hosseini, Sergio O. Martinez-Chapa, Mazhar Sher

**Affiliations:** 1School of Engineering and Sciences, Tecnologico de Monterrey, Ave. Eugenio Garza Sada 2501, Monterrey 64849, Mexico; A00831053@itesm.mx (H.H.); smart@tec.mx (S.O.M.-C.); 2Writing Lab, Institute for the Future of Education, Tecnologico de Monterrey, Monterrey 64849, Mexico; 3Department of Mechanical Engineering and Applied Mechanics, School of Engineering and Applied Science, University of Pennsylvania, Philadelphia, PA 19104, USA

**Keywords:** wearable contact lens, IOP measurement, drug delivery, electrospinning, glucose monitoring, colorblindness

## Abstract

In recent years, wearable contact lenses for medical applications have attracted significant attention, as they enable continuous real-time recording of physiological information via active and noninvasive measurements. These devices play a vital role in continuous monitoring of intraocular pressure (IOP), noninvasive glucose monitoring in diabetes patients, drug delivery for the treatment of ocular illnesses, and colorblindness treatment. In specific, this class of medical devices is rapidly advancing in the area of drug loading and ocular drug release through incorporation of electrospun fibers. The electrospun fiber matrices offer a high surface area, controlled morphology, wettability, biocompatibility, and tunable porosity, which are highly desirable for controlled drug release. This article provides an overview of the advances of contact lens devices in medical applications with a focus on four main applications of these soft wearable devices: (i) IOP measurement and monitoring, (ii) glucose detection, (iii) ocular drug delivery, and (iv) colorblindness treatment. For each category and application, significant challenges and shortcomings of the current devices are thoroughly discussed, and new areas of opportunity are suggested. We also emphasize the role of electrospun fibers, their fabrication methods along with their characteristics, and the integration of diverse fiber types within the structure of the wearable contact lenses for efficient drug loading, in addition to controlled and sustained drug release. This review article also presents relevant statistics on the evolution of medical contact lenses over the last two decades, their strengths, and the future avenues for making the essential transition from clinical trials to real-world applications.

## 1. Introduction

Contact lenses have recently emerged as intelligent health monitoring devices and have been widely utilized in various fields, such as (i) biochemistry and molecular biology, (ii) pharmacology and toxicology, (iii) immunology and microbiology, (iv) neuroscience, and (v) healthcare professions. Figure 1A presents a summary of the statistics on the usage of contact lenses in different research fields, with specific emphasis on the area of medicine. In particular, within the last two decades, contact lenses have found a unique place in the area of biosensing for continuous intraocular pressure (IOP) measurement and monitoring, as well as glucose detection. Contact lenses have also shown great potential as effective ocular drug delivery systems in the treatment of several ocular diseases, including dry eye syndrome (DES), fungal keratitis, and glaucoma, among others [1,2,3,4,5,6]. Figure 1B demonstrates the evolution of the scholarly reports on medical contact lenses in the context of their three main applications, which are also the focus of this review article. These smart devices replace invasive and/or minimally invasive methods of monitoring and treatment of ocular disorders. Hybrid contact lenses incorporate rigid segments that are mounted onto soft lenses that offer comfort and enhanced optical features [7]. While a vast majority of these smart contact lenses are aimed at monitoring IOP changes in the eye over a stretched time period, the interest in this area has dramatically decreased (Figure 1B) over the last 15 years due to the challenges that these systems pose for this specific purpose (thoroughly discussed in Section 5).

In one of the latest reports on IOP measurements, Kouhani et al. (2020) presented a modified planar-doughnut-shaped contact lens integrated with a wireless sensor that is able to monitor continuous changes in the curvature of the cornea [8]. The SENSIMED Triggerfish (^®^) contact lens sensor has been legitimized by the Food and Drug Administration (FDA). This technology introduced an implanted microscale sensor that employs a coil antenna to automatically report the eye curvature changes of the cornea, and transfers the data to an external reader [9]. Another class of contact lenses is targeted at measuring and monitoring other vital elements, including glucose [10], *glaucoma* biomarkers (most commonly, IL-12p70) [11], and ocular pulse amplitude (OPA) [12]. Google and Novartis are the first companies to present the concept of wirelessly powered contact lenses that measure and report the glucose concentrations in the tears of diabetic patients [13,14]. Contact lenses are an excellent candidate for sustained drug delivery as well. The continuous increase in published research articles highlighting advancements of medical contact lenses for drug delivery applications is a testimony of their great potential (Figure 1). In a recent study, a remotely controlled smart contact lens for noninvasive glucose monitoring and subsequent controlled drug release was proposed to treat diabetic retinopathy [15]. In glaucoma patients, application of topical anti-glaucoma drugs is vital for prohibiting the damage to retinal and optical nerves [16]. A recent invention demonstrated embedded microtubes within a contact lens for delivering in situ anti-glaucoma medication. This technology presented fewer side effects and a greater bioavailability than its counterparts while using a considerably lower quantity and concentration of drugs as compared to conventional eye drops [17,18].

When drug delivery is the target application for a contact lens, the design and fabrication of the drug loading matrices and drug release mechanisms can be the most crucial and complex steps. Electrospinning has found wide applications in nano- and micro-scaled fiber production, which are favorable for delivering drugs. Electrospun fibers offer several advantages, including a high surface area to volume ratio, tunable morphology, controlled wettability, biocompatibility, biodegradability, and controlled porosity, which can be utilized for smart drug delivery [19,20,21,22,23,24]. Various factors, such as viscosity, density, and surface tension, play a role in the dimension, morphology, and porosity of the developed fibers. The choice or combination of materials, in turn, plays a vital role in controlling other essential parameters, such as oxygen permeability, biocompatibility, and biodegradability of the drug delivery matrices. For each specific drug, the release rate and profile have to be carefully controlled with respect to the natural optical and diffusive characteristics of corneal tissue to overcome dose management issues and limitations associated with poor delivery [21,22]. Several reports of the literature highlight the design and fabrication of polymer/co-polymer fiber networks for drug delivery applications [20,21,22,25,26,27,28]. Regardless of these advancements, however, several issues are yet to be addressed. The sensitivity of the corneal layer, the negative impact of contact lenses on the visual field during long-term use, the decrease in transparency and drug entrapment, and the slow release rate are some limitations to mention among a number of challenges, which call for further research and improvement in the design and fabrication of these delivery systems.

This article provides a summary of the fundamental developments of the wearable contact lenses from 1967 to date and discusses their role as an appropriate substitute for invasive and/or semi-invasive biomedical devices and gold-standard methods. The current work highlights the advancements of contact lenses in medical applications, offering discussions on the existing smart contact lenses for IOP measurement and monitoring, biomarker detection (in specific, glucose biorecognition for diabetes), colorblindness, and drug delivery while suggesting some windows of opportunity for future research. With a specific emphasis on the role of electrospun fibers in the design and fabrication of drug delivery contact lenses, we critically review pioneering electrospun fiber candidates integrated within the structure of medical contact lenses for controlled drug delivery. Subsequently, the article introduces the fabrication strategy for each class of fibers, applied materials, loading and release mechanisms, and targeted drugs for specific ocular diseases, in addition to some of the advantages and shortcomings of each delivery system. We also highlight the existing challenges that current contact lenses pose in general and those specific to drug delivery applications, including contact lens impermeability to oxygen, eye irritation and discomfort, burst release and changes in drug diffusion rate, and drug entrapment.

## 2. Methodology

In this article, we used the Systematic Reviews and Meta-Analyses (PRISMA) technique as a powerful method for selecting articles that are tightly related to the topic (Figure 2). Using the query strings presented in Appendix A, we collected a total of 20,083 articles and checked for the originality and availability of the documents.

Moreover, we limited our analysis to research articles only and excluded white papers, books and book chapters. Additionally, we narrowed down the application area to IOP measurement, diabetes, drug delivery, and colorblindness. Another applied filter sieved the articles to only consider those that involved human and/or animal studies. As a result, a total of 134 papers were included in this meta-analysis.

## 3. Contact Lenses for IOP Measurement

One of the main contributing factors to ocular illnesses, including glaucoma, is the increase in the pressure within the intraocular structure of the eye. The gold-standard method for IOP measurement is tonometry, which is performed with Goldmann applanation tonometer (GAT). While the GAT method is widely accepted for its accuracy, it can imply certain errors due to corneal thickness and its biomechanical properties [29,30]. Aside from the measurement errors, GAT is complex and semi-invasive, and requires topical anesthesia. The applanator of the device comes in close interaction with the patient’s cornea, which causes discomfort. The above-mentioned process should be performed by clinicians and not by the patient or within a home setup. IOP can vary during diurnal and nocturnal times and with body posture [31,32]. A continuous IOP measurement during daily activities or even throughout sleep can offer invaluable information to prevent patients from entering acute phases of glaucoma. Moreover, the continuous monitoring and measurement of IOP can impact the treatment regimens that clinicians design for individual patients. While traditional IOP measurement falls short of serving patients in continuous measurement and monitoring of the pressure within the intraocular system, soft wearable contact lenses have facilitated this necessary step and provided valuable solutions for continuous IOP monitoring and measurement.

A vast majority of the reported contact lenses are designed for IOP measurement and monitoring. Several reports of effective integrations of contact lenses and biosensors for simultaneous IOP monitoring that include a piezo-resistive effect [1,4,33,34], as well as inductive and capacitive reactance [5,6,35,36], exist within the literature. Appendix A provides a comprehensive summary of the existing contact lenses that have targeted IOP measurement and monitoring. Among the first attempts to develop contact lenses for continuous IOP measurement, in 1974, Green and Gilman proposed a strain gauge attached to a soft contact lens that measured the IOP over a range of 20 to 57 mmHg [37]. Pang et al. (2019) used the Wheatstone bridge circuit to develop a contact lens with the purpose of monitoring and measuring IOP in a non-invasive manner. The structure of the contact lens includes a strain gauge with a metal electrode for recording the fluctuation of the eyeball generated by IOP. The evaluation of this contact lens was conducted on an eye model that represented an excellent and sensitive cycling function at various IOP speeds [3]. Using an electrical resistance–inductance–capacitance resonant circuit, Kim et al. (2017) designed a contact lens to monitor IOP while measuring glucose within tears. Through in vivo and in vitro experiments on male New Zealand rabbits and using a bovine eyeball, the reliable performance of the contact lens was confirmed, which subsequently facilitated the specific binding of the graphene surface to the biomolecules of interest. This technology was found useful, as it could be tuned for a spectrum of target biomolecules, including glucose [10]. Araci et al. (2014) applied microfluidic principles in fabrication of a contact lens that represented a noticeable sensitivity with an accuracy of 1 mmHg in addition to ease of fabrication for mass production. The sensor was an integrated optical device for IOP readout through a camera of a smartphone, which was provided with an optical adaptor and software for analyzing the images. This device specifically targeted alterations in the aqueous–air interface position [38]. Another approach was reported in 2003 by Leonardi et al., in which a microfabricated contact lens with an integrated strain gauge for measurement of IOP was shown to be sensitive enough to measure small variations in cornea’s curvature. The readout was implemented by two active and two passive gauges for double sensitivity and thermal compensation, respectively [33]. The same researchers also proposed an embedded strain gauge soft contact lens in 2004 that measured the alterations cornea’s curvature in a juvenile porcine eye and further correlated the results to IOP variations. This contact lens recorded electrical signals using an embedded resistive gauge with a Wheatstone bridge [1]. In 2009, Leonardi et al. sought to remove the drawbacks of their previous designs of contact lenses by eliminating wires, which posed a discomfort for the user and promoted air bubbles between the cornea and the lens. For this reason, the researchers built a smart contact lens that was wirelessly powered and transferred data between the lens and an outer recording device using an embedded antenna and microprocessor [39]. Laukhin et al. (2011) presented a prototype sensor for a contact lens based on a piezo-resistance mechanism that transferred the fluctuations of cornea curvature to the integrated sensing bi-layer film of the contact lens. This strategy depicted an acceptable range of accuracy in continuous measurement of IOP [4]. A chip-less inductive-coil-based contact lens sensor that followed IOP changes was effectively established and examined by Chen et al. (2014). The evaluation of the sensing apparatus was performed via an external reading coil using an eye model. Using the proposed sensor, the outcomes represented ~8 kHz/mmHg of IOP measurement with an outstanding linearity of R = 0.998. The results showed a high sensitivity of the sensor in tracking the fluctuation of IOP in the silicone model eye [5]. Chen et al. (2009) described a microfabricated sensor for the creation of a foldable and flexible coil substrate using parylene C when implanting for its use in minimally invasive suture-less surgery. The proposed contact lens was tested in vivo and ex vivo in two different sets of experiments, providing a pressure sensing accuracy of 1 mmHg and sensing distance of 2 cm by using an enucleated porcine eye for the validation of device’s feasibility. Moreover, the proposed contact lens complied with surgical requirements, as it projected a strong attachment to the iris and compatibility with the intraocular environment over an extended period of time [36]. Mansouri and Shaarawy (2011) developed a wireless ocular telemetry sensor (OTS) (Sensimed AG, Switzerland) that was used to continuously measure IOP for individuals who suffered from open-angle glaucoma. This contact lens was based on silicone and used a micro-electromechanical system (MEMS) for recording and monitoring IOP-induced corneal curvature alterations. An antenna that was locally mounted around the eye received and transmitted the data to a recorder. The device was tested on 15 patients without any adverse reports, and the OTS showed appropriate function in measuring IOP changes within 24 h [40]. In an interesting report, Song et al. (2019) proposed a smart contact lens capable of measuring IOP, delivering drugs in situ, and detecting biomarkers of glaucoma, all in one contact lens. Contact lens evaluations demonstrated that the sensor could detect Interleukin 12p70 as the glaucoma biomarker in artificial tears on a scale of 2 pg/mL concentration. Additionally, by carrying out experiments on cadaver pig eyes in ex vivo tests, the above-mentioned sensor measured IOP from 10 to 50 mmHg, with an excellent reproducibility [11]. For the first time, a successful contact lens tested on the human eye for continuous measurement of IOP and OPA over 24 h was reported by Wasilewicz et al. (2020) (Figure 1).

The data, including IOP and OPA, were received via a pressure-measuring contact lens (PMCL) device and wirelessly transferred by a periorbital adhesive patch antenna to a recorder. A computer then received the stored recordings to read out and visualize. The results exhibited that the PMCL performed IOP measurements constantly for 24 h in a great number of recordings (325,000 times in day and nighttime) [12]. Lee et al. (2020) produced a contact lens with moiré patterns for measuring IOP changes in a rabbit with acute glaucoma. The integration with an image that was virtually generated from a computer addressed the need for overlapping an additional contact lens, hence eliminating IOP-proportional moiré patterns. In vivo study experiments and temperature-triggered drug elution were tested on white New Zealand rabbits [41]. Campigotto et al. (2019) established a camera inside the microchannel of a contact lens to record the fluid movement, and this was tested on a porcine eye. The developed device was sensitive and offered consistent IOP readouts across several porcine eyes, and the working principle was based on a micro-pressure catheter that was merged into the center of the vitreous chamber [42]. In a recent investigation, Kouhani et al. (2020) presented a wireless passive contact lens that was tested ex vivo on a canine eye, which allowed constant observation of the IOP changes. The sensor was encapsulated entirely within a contact lens with a doughnut shape, which was made of polydimethylsiloxane (PDMS) and was shown to be biocompatible, soft, and penetrable to gas and moisture. The results showed a pressure responsivity of 523 kHz per 1% axial strain on the PDMS membrane and 35.1 kHz per 1 mmHg change in the IOP of a canine eye. In order to eliminate any further predictable effects on the function of the sensor, a real-time calibration for humidity and temperature control was set for the device under appropriate conditions [8]. In 2019, a United States Patent by Araci et al. introduced a contact lens and a closed microfluidic network embedded within the device. The network had a volume sensitive to the applied strain. In this strategy, a liquid–gas equilibrium pressure interface was used to measure the IOP on enucleated porcine eyes to help the regulation of the synaptic plasticity of neurons [43]. Maeng et al. (2020) developed a calorimetric smart contact lens that contained an integrated IOP sensor, which did not require any source power and worked via visual color changes. The in vitro experiments represented the performance of the contact lens on a silicone eye model and a porcine eyeball for ex vivo evaluation with limits of detection of 3.2 and 5.12 mmHg. IOP changes were in direct correlation with color changes of the contact lens, and therefore, by using a cellphone, the color could be easily detected by checking the change in the rigid gas permeable (RGB) values [44]. In another study, Agaoglu et al. (2018) reported a smart contact lens that employed volumetric amplification as a novel transduction mechanism with a passive embedded microfluidic sensor for measuring IOP. It was tested ex vivo on porcine eyes and presented long-term continuous monitoring for more than 19 h and a lifespan of more than 7 months [45]. A non-invasive IOP measurement contact lens was fabricated by Xu et al. (2020) via PDMS spin-coating on a silicon wafer, which included some graphene layers. Two strain gauges in the radial direction and two other compensating resistors on the edge of the sensor were used for the IOP detection. The contact lens demonstrated a remarkable stability, sensitivity, and durability for continuous measurement of IOP over a long term of 24 h [46].

## 4. Contact Lenses for Glucose Detection

Diabetes is known as a metabolic disorder triggered by high blood glucose levels [47], which can be accompanied by vascular damages affecting the heart, eyes, and nerves and resulting in different complications [48]. The global prevalence of this widespread illness has seen a faster pace in the regions of the world that undergo rapid urbanization [49,50,51]. In 1980, the World Health Organization (WHO) reported that a total of 108 million people were dealing with diabetes, a number that rapidly grew four-fold by 2014 [52,53]. The prevalence of diabetes worldwide is estimated to be 4.4% in 2030 for all age categories, growing from 171 million in 2000 to 366 million in 2030 [54]. The most significant demographic change in the prevalence of diabetes worldwide seems to be the surge in the proportion of individuals who are aged above 65 years [54].

A number of studies highlighted noninvasive glucose detection in diabetic patients, and they systematically replaced the conventional and mostly invasive measurement methods (e.g., finger pricking for drawing blood) [55,56,57,58,59,60,61]. While some patients may require from six to eight glucose measurements per day for multiple-dose insulin injections or insulin pump therapy, this test is not frequently performed due to the inconvenience that it involves, especially for children [62,63]. There is a direct correlation between the level of glucose in tears and that in the blood, and it is an excellent indicator for the production of sensor-integrated contact lenses [64]. Such contact lenses are able to identify patients glucose levels from tears and monitor the changes in a non-invasive manner [10,64,65,66]. Biorecognition of tear biomarkers not only facilitates diagnosis and monitoring of ocular diseases (e.g., diabetic retinopathy, glaucoma, dry eye disease (DED), vernal conjunctivitis, Graves’ ophthalmopathy, and ocular tumors) [67,68,69,70,71,72], but also are used for the detection of systemic diseases (e.g., multiple sclerosis, cancer, insulin-dependent diabetes mellitus, and Sjögren’s syndrome) [73,74,75,76,77,78,79]. Within the last three decades, the protein composition of tear fluid has been extensively investigated by using proteomic methods [80], resulting in a rise in the identification of tear proteins from 500 main proteins [81,82] to 1526 in recent studies [67]. The development of smart contact lenses with well-designed tear sampling, storage, and analysis compartments can be a significant step forward in addressing tear biomarker detection. Among different ocular diseases that can be detected via tear biomarkers, almost all reported soft wearable contact lenses have targeted diabetes detection. Appendix A collects the existing literature reports on proposed contact lenses that are either solely dedicated to glucose monitoring or have combined IOP measurement with glucose detection. In 2011, Chu et al. fabricated a new contact lens to measure the level of tear glucose and tested it in vivo. A glucose-oxidase-modified electrode was used on a PDMS contact lens through MEMS technology. Due to the structure of the wearable contact lens, minimum irritation was caused in the eye. The obtained calibration domain included the glucose concentrations within tears in both normal and diabetic patients [64]. Liao et al. (2011) introduced a sensing platform that was embedded into the contact lens and was able to read the glucose level of a tear film wirelessly. The above-mentioned contact lens had an integrated loop antenna to supply the power for data transmission, an integrated circuit (IC) interface of a low-power system, and a wireless sensor for glucose measurement in tears that could be powered from 15 cm of distance. The readout structure was utilized to make a connection between the on-lens light-emitting diode (LED) and contact lens wearer to produce immediate visual feedback [83]. An integrated contact lens with a photonic structure for continuous glucose monitoring was developed by Elsherif et al. (2018). This smartphone-based system benefited from the physiological conditions of the eye, including 150 mM ionic strength and the pH of 7.4 in the eye environment. The fabrication method was rapid (5 min), and the entire process was performed in one single step. This sensor had several beneficial features, including a brief saturation time of 4 min, a noticeably short response period of 3 s, and a great sensitivity of 12 nm mM^−^^1^ [84]. Park et al. (2018) merged smart wearable sensors with soft contact lenses to monitor and record graphene’s resistance change when conjugating to glucose. This integrated system was proposed for remote monitoring of diabetes and functioned based on transferring an electrical signal to the contact lens wirelessly via an antenna. Subsequently, the LED pixel activation prompted the glucose, and the contact lens was tested in vivo, where it successfully carried out real-time glucose monitoring [85]. By using an electrical resistance–inductance–capacitance resonant circuit, Kim et al. (2017) designed a contact lens to monitor glucose within tears and to measure and screen IOP. In vivo and in vitro studies were performed on live male New Zealand rabbit and bovine eyeballs [10]. In a recent work, Kim et al. (2020) developed a glucose biosensing method whereby the device provided a real-time glucose measurement in the tear sample of a diabetic rabbit’s eye and an on-demand drug administration targeting diabetic retinopathy (Figure 2). A considerable level of sensitivity, linearity, and stability was recorded for this ocular delivery device over the course of 63 days. The genistein delivery through the cornea to the retina was proven to have a comparable therapeutic impact to that of the intravitreal injection of Avastin [15]. Recently, Guo et al. (2020) developed a facile method for fabricating a multifunctional contact lens with ultrathin MoS2 transistors and Au wire sensors. The contact lens was shown to have the capability of monitoring glucose directly from tear fluid, receiving optical information through a photodetector, and detecting corneal disease with a temperature sensor. In vitro studies presented its high detection sensitivity, and the device was proven to be fully nontoxic and biocompatible [86]. Moreddu et al. (2020) integrated paper microfluidics within a commercial contact lens that had the capability of simultaneous evaluation of biomarkers/proxies, including glucose, proteins, ascorbic acid, nitrite ions, and pH. The received data were collected, stored, and analyzed using a smartphone application. The paper’s functionality of detecting analytes was reported to be within 2 μL of artificial tear fluid over a period of 35 s. Additionally, the contact lens and sensors demonstrated acceptable durability via exposure to UV light for up to 30 min [87]. In 2017, Ruan et al. developed a novel glucose sensor with a crystalline colloidal array (CCA) integrated into a hydrogel matrix placed on a RGP contact lens. Changes in glucose concentration resulted in diffraction of the light wavelength, which acted as a clear indicator. Hence, the sensor functioned by selectively identifying such diffractions. The contact lens was able to bind specifically to glucose in the presence of other analytes in simulated tear fluid (STF), which resulted the response of the sensor at around 10 nm. This sensor could diffract the visible light between 567 and 468 nm to reddish yellow, green, and blue [88].

## 5. Contact Lenses for Colorblindness

Three kinds of retinal photoreceptors are necessary for humans’ normal color vision, which is also known as normal trichromacy. These photoreceptor cones—namely, the short (S) cone, medium (M) cone, and long (L) cone are situated at the back of the eyes. Each cone has a specific spectral response based on the particular type of photopigment it includes. The above-mentioned cones are also exclusively sensitive to colors—for instance, the S cone to blue, M cone to green, and L cone to red [89]. Moreover, according to the received wavelength, the cones can be activated and can combine the colors at different levels to produce color signals. Colors play a crucial role in understanding our surroundings and influencing our emotions. It can therefore result in a range of challenges if colors are not precisely observed [90].

People with normal color vision are referred to as trichromats. Colorblindness, or color vision deficiency (CVD), is an ocular disease whereby patients are not able to match or discriminate between colors. People who suffer from CVD are usually described by their kind of deficiency. Based on the defect type, there are three CVD groups, including dichromacy (missing photoreceptor cone), anomalous trichromacy (faulty photoreceptor cone), and monochromacy (at least two missing photoreceptor cones) [91,92]. In recent years, extensive experiments have been conducted for CVD treatment. Despite the studies in the field, most CVD patients still rely on wearables, including tinted glasses or lenses, to overcome the daily difficulties of their routine tasks [93,94]. As one of the renowned producers of tinted glasses, the Enchroma company provides glasses for CVD sufferers [95], and recent smart glasses for CVD patients fabricated by Google can help in overcoming the challenges of colorblindness [96,97]. While such wearables are used to actively filter and recolor the vision of the CVD patients by employing image-processing algorithms, the current color-corrective glasses are rather large in size (impractical for daily use), expensive, and incompatible with other vision correction lenses.

Smart lenses were introduced to offer higher efficacy, stability, and ease of wear. Appendix A gathers the existing literature regarding contact lenses for colorblindness. Companies such as Chromagen have presented red contact lenses to help CVD patients [98]. Recently, in 2021, Salih et al. fabricated a color-filtering contact lens in which gold nanoparticles were embedded for creating nanocomposite contact lenses for red–green CVD patients. Three distinct sets of nanoparticles, in addition to hydroxyethyl methacrylate (HEMA), were used as a base polymer via a cross-linker named ethylene glycol dimethacrylate (EGDMA). The gold nanoparticles filtered and differentiated a variety of optical wavelengths that CVD patients were barely able to distinguish. Overall, the developed contact lens showed superior water retention and wettability characteristics in comparison to available commercial CVD wearables. In another study by Salih et al. (2021), spherical silver nanoparticles (SNP) were embedded into HEMA-EGDMA hydrogel contact lenses for blue–yellow CVD patients with great material and optical features. The contact lens was able to filter more than 65% of light in the 488 and 494 nm wavelengths. The known properties of spherical silver nanoparticles were leveraged to attract visible light in the 390–490 nm range to help blue–yellow CVD patients. Additionally, the cytobiocompatability analysis represented a good biocompatibility for the contact lens and presented it as an excellent candidate for utilization in clinical trials. A color-filtering contact lens was developed by Badawy et al. (2018) to filter out specific light wavelengths of around 545–575 nm through the incorporation of rhodamine dye in a commercial contact lens. After 72 h of testing the device, the contact lens indicated no toxicity and 99% cell viability for human corneal epithelial cells. By using the dip-and-drop method, a drop of the dye was cast on the lens surface and subsequently dipped in the dye solutions for 1 min to create a suitable contact lens for CVD management.

## 6. Contact Lenses for Drug Delivery

Eyedrops of relevant drugs are widely utilized to treat various eye diseases. Methods that are used for drug delivery and treatment of ocular disorders are significantly different based on the nature and level of the illness. According to the sophisticated structure of the eye, there are some unique challenges to be addressed by drug delivery strategies. Ocular diseases could be related to tissues at the back of the eye (BOTE), including diabetic retinopathy and age-related macular degeneration, or could be in relationship with the front of the eye (FOTE), which significantly impacts the drug delivery approaches. Procedures of drug delivery for subcutaneous or BOTE delivery could be injections and sustained-release implants; meanwhile, there could be a higher risk of contamination, retinal damage, and/or internal ocular bleeding [99].

The ophthalmic drugs that are administrated through existing topical eye drops have proven to have limited ocular bioavailability, as the physiological structure of the eye, in its anatomical nature, is somewhat complex and unique [100,101,102,103,104]. Mostly, eyedrops act in a short time, which leads to only 1–7% of the dosage reaching the desired area, as the eye’s movements clean off the rest. Moreover, there are two forms of drugs—named gutta and oculentum—that, though are used 60% more than other drug types, have proven to have a poor bioavailability [105]. Furthermore, the physiological and anatomical structure of the eye makes ocular drug delivery a complex task, and a specific drug concentration released at a particular site can be a serious concern. Therefore, less than 1% of topically administered ophthalmic drugs are able to reach the aqueous surface of the eye [11].

Inevitably, a new system of drug delivery is needed for the expansion of drugs’ bioavailability and increased adherence of patients in order to have better medical results. Delivering drug components via contact lenses has attracted considerable attention due to the desirable features of such devices, including easy removal of contact lens—hence, termination of therapy—extended wearing time, and more than 50% bioavailability in comparison to eyedrops [106,107]. To attain sustained drug delivery and to load ophthalmic drugs into the matrices of contact lenses, different reports have proposed drug–polymeric film, soaking methods, imprinting, and drug-laden nanoparticles [108,109]. A considerable advancement in the field has been made, while some issues, including fluctuations in the optical and swelling properties and high burst release remain unsolved [110,111].

Reports in the literature have provided new methods of delivery by novel contact lenses (Appendix A). Maulvi et al. (2017) fabricated a hyaluronic acid (HA)-laden ring that was proposed for an implantable contact lens. This device was made via a modified cast-molding technique to avoid critical alterations in the properties of the contact lens. The contact lens was aimed at increasing the usage time for the ocular residence of HA while facilitating a sustained ocular delivery for DES treatment (Figure 3). An in vivo pharmacokinetic evaluation was performed using rabbit tear fluid, and the results demonstrated a controlled HA release over the course of 15 days [112].

Mun et al. (2019) developed a micelle-embedded contact lens with cholesterol–hyaluronate (C–HA) to extend the time of drug delivery and provide an efficient hydrophobic drug loading. The fabrication of the contact lens was conducted through HEMA via the cross-linker EGDMA. The integrated C–HA micelle contact lens not only continued the optical transmittance, but also represented an excellent improvement of 77.55° in the wettability. Additionally, in vitro tests for drug delivery of cyclosporine showed an organized delivery for more than 12 days [113]. In a different study, Huang et al. (2016) presented a contact lens based on hybrid hydrogel, where they integrated silver nanoparticles, quaternized chitosan, and graphene oxide (GO) charged with an antifungal and antibacterial blend within its structure. The contact lens delivered Voriconazole (Vor), an antifungal component, and improved its continuous delivery from the hydrogel network and maintained the drug release. The contact lens was tested in vitro and in vivo and exhibited acceptable antimicrobial functions (Figure 4).

The Vor-loaded hydrogel-based contact lens proved to have excellent in vitro efficacy in antifungal activity while enhancing the therapeutic effects when tested on a fungus-infected mouse model [105]. Recently, Yan et al. (2020) used molecular imprinting to increase the contact lens’s loading with sustained release efficiency of bimatoprost. The conventional soaking strategy was used for bimatoprost loading, and the results were compared with those of molecularly imprinted contact lenses. The device increased the uptake of bimatoprost from the main solution in rabbit tear fluid and also improved its release kinetics from the lens matrix by around 28.22 ± 3.33 μg [114]. Song et al. (2019) proposed a contact lens with three different functions, including IOP measurement, in situ drug delivery, and glaucoma biomarker (IL-12p70) detection in artificial tears. The contact lens was also aimed at constant drug delivery for a minimum of 30 days and was found to be capable of monitoring IOP with excellent repeatability. In order to reduce the cost and simplify the process of fabrication, all functional parts of the contact lens were fabricated from anodic aluminum oxide thin film and optically transparent materials. Experiments proved that a sustained drug delivery was achievable by implementing the drug depot methodology and nanopores [11]. Lee et al. (2019) designed a bicontinuous microemulsion contact lens (BMCL) loaded with thermosensitive poly(*N*-isopropyl acrylamide) nanogel of timolol maleate (TM). The BMCLs could be triggered to release TM according to body temperature. The daily delivery of the drug was controlled via the loading technique for continuous drug delivery and the initial volume of the loaded drug, which decreased the drug loss during storage or transportation [115]. Desai et al. (2020) proposed multiple implant-laden contact lenses for passive delivery of TM, HA, and bimatoprost. The above-mentioned contact lens was targeted at delivering therapeutically relevant doses of these drugs and simultaneously removing the risk of high-burst drug delivery. The loading compartments were separately embedded in the external edge of the silicone-based contact lenses. The proposed contact lens was free from preservatives, and demonstrated successful performance in preventing initial burst release and its subsequent side effects while improving patient’s comfort and facilitating the administration of multiple therapeutics. The proposed contact lens family is an excellent candidate for the treatment of chronic diseases, such as glaucoma [116]. In another study, Ran et al. (2020) developed a contact lens with an embedded sparfloxacin-laden ring that was coated with polyvinyl pyrrolidone (PVP) for sustained ocular drug delivery without affecting the main optical and swelling features. All of the batches showed a sustained drug release for up to 48 h in vitro and in vivo [117]. Using the same polymer, Xue et al. (2020) fabricated a doughnut-shaped contact lens with PVP-coated olopatadine-ethyl cellulose microparticles laden for the same purpose as Ran et al. A modified casting technique was used to integrate the doughnut segment within the lens’s periphery. In vivo tests were conducted on the tear fluid of a rabbit. A considerable improvement in the retention time (~48 h) for olopatadine HCl, and PVP was recorded when compared with that of conventional eyedrops [118]. ElShaer et al. (2016) developed a polymeric drug delivery system that addressed several crucial factors, including stabilizers, polymer mixture composition, and the amount of active prednisolone for delivery. For such a target, via a single-emulsion solvent evaporation technique, poly(lactic-co-glycolic acid) (PLGA) nanoparticles with an average size of 347 nm were prepared and incorporated into a transparent contact lens. In vitro drug delivery experiments showed a continuous and prolonged drug release over a period of 24 h [119]. In another study in 2018, a molecularly imprinted color contact lens for continuously delivering TM was developed by Deng et al. This therapeutic contact lens could use colorimetric assays for analysis of TM release for self-reporting using color changes. The sensor mechanism permitted the effective translation of the drug delivery into a readable signal by its color change. Moreover, the functional groups incorporated into the hydrogel matrix resulted in color changes in the contact lens through the molecular stimulation of the surrounding environment. TM release was tested in an artificial tear fluid, and the level of TM delivery was visually checked by using the visible color change in the lens [120]. Wang et al. (2016) proposed a thin micropump-integrated contact lens to deliver drugs without battery power. The developed system was actuated via an external magnetic field of 152 to 469 mT that could precisely control the drug delivery at the nanoliter scale on demand. The PDMS membrane was used as the main component, while a magnetic nanoparticle–PDMS composite (MNPC) was employed for the actuation under the magnetic field pulses. The final thickness of the sensor was less than 500 μm, which made it compatible for integration into a contact lens for drug delivery purposes [121].

## 7. Electrospun-Fiber-Incorporated Contact Lenses for Drug Delivery

Since the advent of electrospinning approximately three decades ago, this method of fabricating fibers has garnered considerable attention in the production of fibers that range from micro to nano scales [122,123]. Electrospun fibers are produced from diverse materials, including conductive, synthetic, and natural polymers, composites, ceramics, or even metals [124,125,126,127]. In specific, electrospun fibers have proven to be excellent candidates for drug delivery systems [28,128,129,130,131,132,133,134]. Compared to other matrices for drug delivery, electrospun fibers have been shown to have several advantages, including porosity and a high surface-area-to-volume ratio, which can be used for effective drug loading [19,20,21,22]. Moreover, the loading and delivery of drugs can be manipulated with modifications made in the fiber membrane’s composition, morphology, and wettability [23]. Even a very slim diffusion passage length can be designed and fabricated with the small dimension of nanofibers [24]. Additionally, fibers can be cost-effective and biocompatible candidates for fabricating drug barriers. In particular, when ocular illnesses are concerned, there is a shortage in bioavailability. Electrospun-fiber-integrated contact lenses make remarkable candidates for ocular drug delivery.

Several attempts in recent years have demonstrated the successful use of electrospun fibers in drug loading and ocular drug release within the frame of wearable contact lenses (Appendix A). Mehta et al. (2017) presented a contact lens with four permeation enhancers (Pes) and performed an in vitro analysis of the TM release profile from the polymeric electrospun nanofibers. The exterior of the contact lens was coated by electrospinning. Some factors were used during the process of electrospinning, including a new system with a electrohydrodynamic (EHD) deposition process, which was used to develop four lenses simultaneously, injection rates between 8 and 15 μL/min, a ground electrode for changing the atomized structures, and a masked template (Figure 5A,B). The experiments emphasized the production of a lens that could be used throughout the day and taken off at night. The structure of the proposed contact lens eliminated multiple barriers, including dosing issues and restrictions related to poor ophthalmic bioavailability of drugs (Figure 5C,D). As a result, the PEs successfully increased drug absorption, thus improving the ocular bioavailability [22].

In 2009, Fuerst et al. proposed a contact lens with an increased refractive correction that not only increased the patient’s comfort, but also increased the lipid and protein deposition for pharmaceutical drug delivery. The above-mentioned contact lens unraveled a new method of fabricating polymer fibril diameters and spacing that duplicated natural corneal collagen’s optical transmission and diffusion characteristics. In this study, fibrils of natural human corneal stromal collagen were generated and covered to shape the basis of a fiber mat that exhibited corneal stromal tissue’s transparency and diffusion characteristics. The electrospinning process involved a direct current (DC) potential of 4000–12,000 volts for the fabrication of collagen fibers. The accumulation of the fiber strands created a polymer mat with diameters ranging from micro to nano scales [20]. In a newly published work, Göttel et al. (2020) fabricated a new solid in situ gelling system to treat topical ocular diseases. In this method, a defined curvature of the sample was made by forming the electrospun fibers with a 3D-printed matrix. Nanofibers with a high porosity and narrow size distribution were produced by electrospinning pullulan–gellan gum solutions. With its tuned geometry, this in situ gelling system was an excellent candidate for ocular drug delivery. The electrospun fibers were made of 20% pullulan and 0.225% gellan gum, and were subsequently shaped into a lens of 2.54 mm ± 0.38 mm height. The curvature-forming process reduced the lens diameter by 1 mm and to 1.4 cm [25]. Mehta et al. (2019) demonstrated that electrospinning could effectively modify the structure of polymers to control the TM delivery and likely to improve the drug permeation without endangering corneal tolerability. The combination of chitosan and atomized coatings could generate particulate structures. The contact lens was tested in vitro, and the results showed a highly controlled performance in glaucoma delivery in an elderly community [26]. In a different study, Mehta et al. (2017) proposed a novel technique using PVP and poly(*N*-isopropylacrylamide) (PNIPAM) polymers with a one-needle electrohydrodynamic (EHD) engineering process for developing nano-scaled coatings on different contact lenses’ surfaces, thus rendering stationary constructs on both the lens and the bio-interface. The process of coating was conducted through an emitted charged droplet, and the fiber optimization controlled fiber permeation, and release kinetics were tested using TM for the treatment of glaucoma (Figure 6A,B). In vitro studies presented a double-phased performance of the coatings when releasing the target drugs. An initial burst was observed, which was followed by a rather sustained release over time. Moreover, TM-loaded poly(*N*-isopropylacrylamide) (PNIPAM) coatings have presented the highest drug dose release (89.8%) after 24 h [28]. Davis et al. (2018) described a patented contact lens that included polymer particles to offer simultaneous lipid permeability and impermeability while offering high oxygen permeability for drug delivery (Appendix A). A proprietary polymer mat was produced by electrospinning a prepolymer solution and was incorporated into a polymeric contact lens. The device exhibited the natural optical and diffusive characteristics of corneal tissue [21]. By using a special mixing ratio of permeation enhancers and polymers, Mehta et al. (2018) presented the capability of utilizing electrohydrodynamic atomization (EHDA) in engineering strong coverings to improve the permeation of pharmaceuticals, thus increasing the bioavailability of ocular drugs (Appendix A). Their ex vivo analyses indicated that by adding permeation enhancers, a substantial increase in TM permeation in comparison to an additive-free setup could be achieved (53.39 ± 3.95 μg cm^−^^2^) after 24 h. This significant improvement in drug permeation resulted in a decline in ocular toxicity due to the lower systematic absorption over an appropriate time frame [27].

## 8. Limitations and Existing Challenges of the Contact Lenses

Despite the significant advancements in contact lenses for IOP measurement and monitoring, glucose detection, colorblindness, and drug delivery, each category still suffers from certain shortcomings. When IOP measurement by contact lenses is concerned, the problem of optically measuring the pressure over a hazy cornea, needing a surgical procedure to implant contact lenses within the eyes, and carrying connectors for electrical readouts are a few from a number of challenges. Most contact lenses developed for IOP measurement and monitoring operate based on capacitive, inductive, or piezo-resistive principles. In all of the mentioned strategies, the contact lens is connected to devices to generate a signal or record the readout measurement via proper wiring, which poses a specific challenge for their application in the real eye. A fully integrated contact lens based on electrical stimulation, therefore, requires the transition to complex flexible electronics and wireless communication, which are yet to be achieved. Moreover, contact lenses that have integrated electronic components commonly present the following shortcomings: (i) The patient’s vision might be compromised due to the application of embedded circuits and electronic materials that are opaque, as well as metal antennas, which can partially or fully block the vision of the user [6,64,135]; (ii) the embedded electronics within the plastic contact lens create deformations upon bending of the material to fabricate curve-shaped contact lens. This may result in sharp pieces that can irritate and damage the cornea, as well as the eyelid [83,136,137,138]; (iii) the signal transmission devices are rather bulky and expensive and limit patients’ activities [1,10,64,83].

Contact lenses that are impermeable to oxygen cause eye irritation and discomfort and project limited hydration properties that are not optimal for tear collection. Soft contact lenses have overcome parts of these limitations by increasing the oxygen permeability and hydration properties [65]. Poly(2-hydroxyethyl methacrylate) (pHEMA) was one of the first polymers introduced for the development of soft contact lenses to receive FDA approval [139]. In 1999, the use of PDMS in the fabrication of contact lenses opened a great window of possibility for more oxygen permeability and patient comfort [140]. In the case of therapeutic contact lenses and drug delivery systems, consumers have described pink eye syndrome, indicating that the developed lenses still pose certain challenges with respect to their biocompatibility with human eyes [141,142]. Other crucial issues, including high-burst release and swelling of the drug-loaded network, are problems that are yet to be tackled [110,111].

Electrospinning presents a number of beneficial features, making electrospun fibers great candidates for ocular drug delivery. Scientists have created a range of fiber networks to focus on both drug delivery and the engineering of corneal tissue. One of the main challenges when dealing with electrospun fibers for drug loading and release is the fine-tuning of the fiber matrix and its morphology, porosity, and wettability, as drug particles may get trapped within the network of fibers, which may act as barriers to hinder the drug diffusion instead of promoting it [22,26,27]. Furthermore, the low efficacy and systemic side effects of these electrospun release systems are additional aspects to be thoroughly studied [143]. Other reports indicated problems, including small changes in a drug’s diffusion rate, that may result in unclear sight due to the use of a contact lens [20]. Apart from biocompatibility and a sustained drug release profile, the fibers’ biodegradability is also of vital importance. The inconsistent expansion of the fiber network in the contact lens can, in turn, pose serious problems [21]. In conclusion, with the fast-growing biotechnological solutions, it is possible to integrate advanced electrospun fiber designs within the structure of contact lenses to partially or fully overcome the existing challenges and tune electrospun-based engineered contact lenses for wider clinical applications. Nevertheless, important aspects, including careful control over the fiber matrix, fiber morphology, porosity, wettability, biocompatibility, and biodegradability, as well as the loading and release profile, are essential topics for further in-depth investigations.

## 9. Conclusions and Future Prospects

Within the last two decades, microfabrication technologies have facilitated the manufacturing of different types of contact lenses. Such soft wearable devices are mainly aimed at the continuous measurement of IOP, glucose detection, colorblindness, and ocular drug delivery applications. Despite the significant breakthroughs in the development of soft wearable medical contact lenses, the reported issues, including eye irritations, patient discomfort, blocking of users’ vision, pink eye syndrome, drug burst, and inconsistencies in drug release, in addition to the high costs of the devices, present an obvious need for further advancement of the field. In this meta-analysis, we review existing reports in the literature on wearable contact lenses for medical applications, with a specific emphasis on IOP measurement and monitoring, glucose detection, colorblindness, therapeutic contact lenses for controlled drug release, and coupled devices that serve more than one purpose. Moreover, we provide an overview of electrospun-fiber-integrated contact lenses, which have shown promise for effective drug loading and sustained and controlled drug release. For each class of contact lenses and their respective applications, we critically review the shortcomings and the current challenges of these devices that are yet to be addressed.

While the timeline of the advancements of wearable medical devices has marked a significant milestone for contact lenses, further research is required to increase the likelihood of the acceptance of such devices by the medical community. Biosensor-integrated contact lenses that rely on tear sample collection and analysis are essential components of these devices. While the scope of analysis is expanding, the quantity of biomarkers that exist within tears may outnumber the biosensors that can be possibly accommodated within a single lens [144]. Multi-target biosensors for medical contact lenses can open windows of opportunity for addressing this problem. This is particularly relevant when a contact lens is coupled with drug compartments for on-demand delivery. The blend of a multiple-analyte sensor and a feedback-based drug delivery system that act in an automated manner can revolutionize personalized points of care [145]. Artificial intelligence can be a valuable addition to the study of the therapeutic effects of these smart devices, while machine learning algorithms can provide long-term monitoring of an individual’s health status, thus further improving the course of treatment [146]. Contact lenses can potentially offer the technical and material novelties required for the next generation of precision devices for medical applications [147]. Moreover, combining medical contact lenses with those used as fashion accessories would promote their use in daily life. The extended time of use can be highly beneficial for various analyses that require daily or overnight wear schedules [65]. Apart from sensors or integrated drug reservoirs, medical contact lenses will also see a great deal of improvement in the choice of novel materials (e.g., graphene, quantum dots, plasmonic nanoparticles, liquid crystals, and bioinspired photonic structures) and in the fabrication strategies applied to create such smart devices [65]. Under any heading, however, contact lenses for medical applications will require extensive validation in human studies and a more profound knowledge of the clinical relevancy to be further embraced by clinicians [148]. This review article offers a fresh look into the advances of marketed and laboratory-based contact lenses, their major applications, and their limitations, while it is a call for further research on unaddressed areas of these soft wearable medical devices.

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
