# Peer review of "A Meta-Analysis of Wearable Contact Lenses for Medical Applications: Role of Electrospun Fiber for Drug Delivery"

_polymers, 2022, doi:10.3390/polym14010185_

Round 1

Reviewer 1 Report

The manuscript by Hosseinian et al. represents meta-analysis on wearable contact lenses for medical applications focusing on the role of electrospun fiber for drug delivery. This analysis is timely while the paper is well written and easy to follow. I have several suggestions only, which I hope will help authors improving the manuscript:

  • In my opinion an Introduction is the wrong place for the description of the methodology of the study. I would like to suggest authors including a special paragraph focusing on it;
  • I think that it would be very important for the reader to know authors opinion on the most promising directions of further development of the area. Therefore, I would like to suggest authors to expand the Conclusions including Further Directions;
  • The paragraph numbering should be corrected;
  • I think that the tables should be moved into the supplementary materials.    

Author Response

Please find our detailed response in the attached document. 

Reviewer 2 Report

The authors realized a comprehensive review about the “Wearable Contact Lenses for Medical Applications”. In my opinion the present review touches all the key points of the presented topic. Before publication I have only a few comments:

  1. Table of abbreviation: for SiO2 the “silica” is more often used than “Silicon dioxide”.
  2. For tables 1 to 5 is better to use landscape orientation of the page, in present form they are hardly understandable and some of the information are missing.
  3. Annex 1 should be moved in supplementary materials.

Author Response

Please find the details in the attached document. 
